# Hot Deformation Behavior of Alloy AA7003 with Different Zn/Mg Ratios

**Xu Zheng** [1,2]**, Jianguo Tang** [1,3]**, Li Wan** [1]**, Yan Zhao** [1]**, Chuanrong Jiao** [1] **and Yong Zhang** [1,3,]*

1    School of Materials Science and Engineering, Central South University, Changsha 410083, China
2    Guangxi Key Laboratory of Materials and Processes of Aluminum Alloys, ALG Aluminium Inc.,
      Nanning 530031, China
3    Key Laboratory of Non-Ferrous Metals Science and Engineering, Ministry of Education,
      Changsha 410083, China
*    Correspondence: yong.zhang@csu.edu.cn

**Abstract:** The hot-deformation behavior of three medium-strength Al-Zn-Mg alloys with different Zn/Mg ratios was studied using isothermal-deformation compression tests; the true strain and true stress were recorded for constructing series-processing maps. A few constitutive equations describe the relationship between flow stress and hot-working parameters. The microstructures were characterized using an electron backscatter diffraction (EBSD) detector and transmission electron microscope (TEM). The results show that the optimized deformation parameters for ternary alloy AA7003 are within a temperature range of 653 K to 813 K and with strain rates lower than 0.3 $S^{-1}$. The microstructures show that materials with a lower Zn/Mg ratio of 6.3 could lead to a problematic hot-deformation capability. Alloys with a higher Zn/Mg ratio of 10.8 exhibited better workability than lower Zn/Mg ratios. The $Al_3Zr$ dispersoids are effective in inhibiting the recrystallization for alloy AA7003, and the Zn/Mg ratios could potentially affect the drag force of the dispersoids.

**Keywords:** deformation maps; Al-Zn-Mg alloys; $Al_3Zr$ dispersoids; Zn/Mg ratios; recrystallization





## 1. Introduction

Al-Zn-Mg ternary alloys have strong work-hardening capabilities at room temperature. In order to achieve the desired microstructures and mechanical properties, it is required that they be processed at elevated temperatures [1,2]. An appropriate thermal–mechanical-processing (TMP) route should be carefully selected, as dynamic recrystallization or cracking may happen during the deformation process.

Prasad developed dynamic material modeling (DMM) to calculate the processing maps using a set of flow stress data as a function of temperatures and strain rates over a wide strain range [3–6]. The calculated processing maps can optimize the hot-processing parameters and determine flow instability regimes that should be avoided during processing. Several published papers have demonstrated that the processing maps have been successfully applied for steels [7], zirconium alloys [8], and aluminum alloys [9–11].

For example, Lin et al. built up the hot deformation and processing map for a typical aluminum alloy AA7075. They proposed that the optimum hot-working domain for this high-strength alloy should be within the temperature range of 623–723 K and strain rate range of 0.001–0.05 $S^{-1}$ [12]. Xiao et al. also demonstrated similar optimum processing parameters for their studied alloy AA7050 [11]. However, Lu et al. also showed that the optimal hot-working processing parameters for alloy AA7075 sheet are within the temperature range of 695–723 K and the strain rate range of 0.05–1 $S^{-1}$ [13]. Their strain rate conditions for a given alloy are different. Zhao et al. claimed that the initial structures for Al-Zn-Mg-Cu before deformation could cause a significant difference. Their results demonstrated that the recrystallization mechanism might be different for different grains. Therefore, the alloys with different microstructures should be deformed accordingly [14]. Luo et al.

studied the deformation behavior of alloy 7A09 during the isothermal-compression test. Their results showed that the maximum power dissipation efficiency was about 0.34 for the studied alloy deformed at 713 K and a strain rate of 0.01 $S^{-1}$ [15]. In comparison, Liu studied the isothermal-compression process of alloy AA7085. Their results demonstrated that dynamic recrystallization could happen if the alloy deformed at a temperature higher than 673 K with higher-strain-rate conditions. Therefore, the alloy should be deformed at a temperature of 673 K and a strain rate of 1 $S^{-1}$ [16]. Yang et al. demonstrated that the optimized deformation parameters for alloy AA7085 are within a temperature range of 663–723 K and at strain rates lower than 0.1 $S^{-1}$ [10].

However, Bylya et al. showed validation of the simulation results compared with processing maps for alloy AA2099. The author claimed that the underlying mechanisms of instability regions for processing maps are unclear. Therefore, more meaningful processing maps might be generated using more complex testing scenarios [17].

In this article, we develop processing maps for alloy AA7003 with different Zn/Mg ratios. This alloy is known as a medium-strength alloy, and it can be easily processed during manufacturing. M. Kumar et al. demonstrated that a medium-strength alloy AA7020 exhibited the desired workability at temperatures above 423 K and was sensitive to temperature and strain rate [18–20]. However, there is always the requirement, from an industrial point of view, that the alloy be extruded as fast as possible. Then, the question would be whether the alloy AA7003 can be deformed at a relatively faster or lower temperature range? Can they be easily deformed with slight change in alloying compositions? It is also reported that such medium-strength Al-Zn-Mg alloys suffer a strong natural-ageing effect after quenching [21–23]. By altering the Zn/Mg ratios, the natural ageing effect can be inhibited, but on the other side, the deformation capabilities for different Zn/Mg ratios should be evaluated systematically. The microstructure characterization of a series of processing maps can indicate different workability of variable alloy compositions.

## 2. Materials and Experiments

The materials used in the present study were deliberately designed AA7003 alloys with three different Zn/Mg ratios, but the total content of Mg + Zn was the same (~6.6 wt.%). The measured chemical compositions are shown in Table 1.

**Table 1.** Chemical composition of the tested 7003 alloys (wt.%).

| Fe | Si | Zn | Mg | Cu | Mn | Ti | Zr | Al | Zn/Mg |
|------|------|------|------|------|------|------|------|------|-------|
| 0.14 | 0.06 | 5.73 | 0.91 | 0.16 | 0.02 | 0.03 | 0.20 | Bal. | 6.3 |
| 0.14 | 0.04 | 5.95 | 0.72 | 0.17 | 0.02 | 0.03 | 0.19 | Bal. | 8.3 |
| 0.14 | 0.05 | 6.05 | 0.56 | 0.18 | 0.03 | 0.03 | 0.23 | Bal. | 10.8 |

The cylinder samples with a dimension of Ø10 × 12 mm were cut along the longitude direction of commercially direct-chilled casting ingots. The homogenization was carried out at 733 K for 48 h. The isothermal-deformation compression tests were carried out on a computer servo-controlled Gleeble-1500 thermo-simulation machine. All samples were lubricated with graphite paste at both ends to reduce friction and increase thermal conductivity. The samples were heated to target temperatures at a constant heating rate of 1 K/s and held at setting temperatures for 5 min before compression. Five different temperatures (653, 693, 733, 773, and 813 K) were chosen to cover the real industrial-manufacturing situations. Samples were deformed at constant true strain rates of 0.01, 0.1, 1, and 10 $S^{-1}$ over a selected temperature range. All samples were deformed to a strain of about 0.9. True stress–true strain curves were recorded during the compression test.

Microstructures were characterized using a ZEISS EVOMA10 scanning electron microscope with an OXFORD electron backscatter diffraction (EBSD) detector. Samples were electrolytically polished in a 10% perchloric acid solution mixed with 90% ethanol. Transmission electron microscope (TEM) samples were 3 mm discs punched from an 80 μm-thick foil. The specimens were further polished using twin-jet electropolishing in a solution of

80% methanol and 20% nitric acid at a temperature below 248 K. A Tecnai G2 F20 TEM then examined the samples operated at 200 kV.

## 3. Results and Discussion

### 3.1. True Stress–Strain Curves

Before the stress–strain curves are presented, the initial microstructures are shown in Figure 1. It is shown that all three alloys demonstrated large grains with casting dendrites inside. The grain size is very close for the alloy with a Zn/Mg ratio of 6.3 (as shown in Figure 1a) and a Zn/Mg ratio of 8.3 (as shown in Figure 1b). The grain size for the alloy with a Zn/Mg ratio of 6.3 is relatively more prominent than the other two alloys. The measured average grain sizes are $200 \pm 35$ μm, $156 \pm 28$ μm, $175 \pm 40$ μm, respectively. However, this is not conclusive as the as-cast microstructure varies from place to place. It is also noticed that some eutectic phases can be observed. This could be due to the high alloying content for the studied alloys.

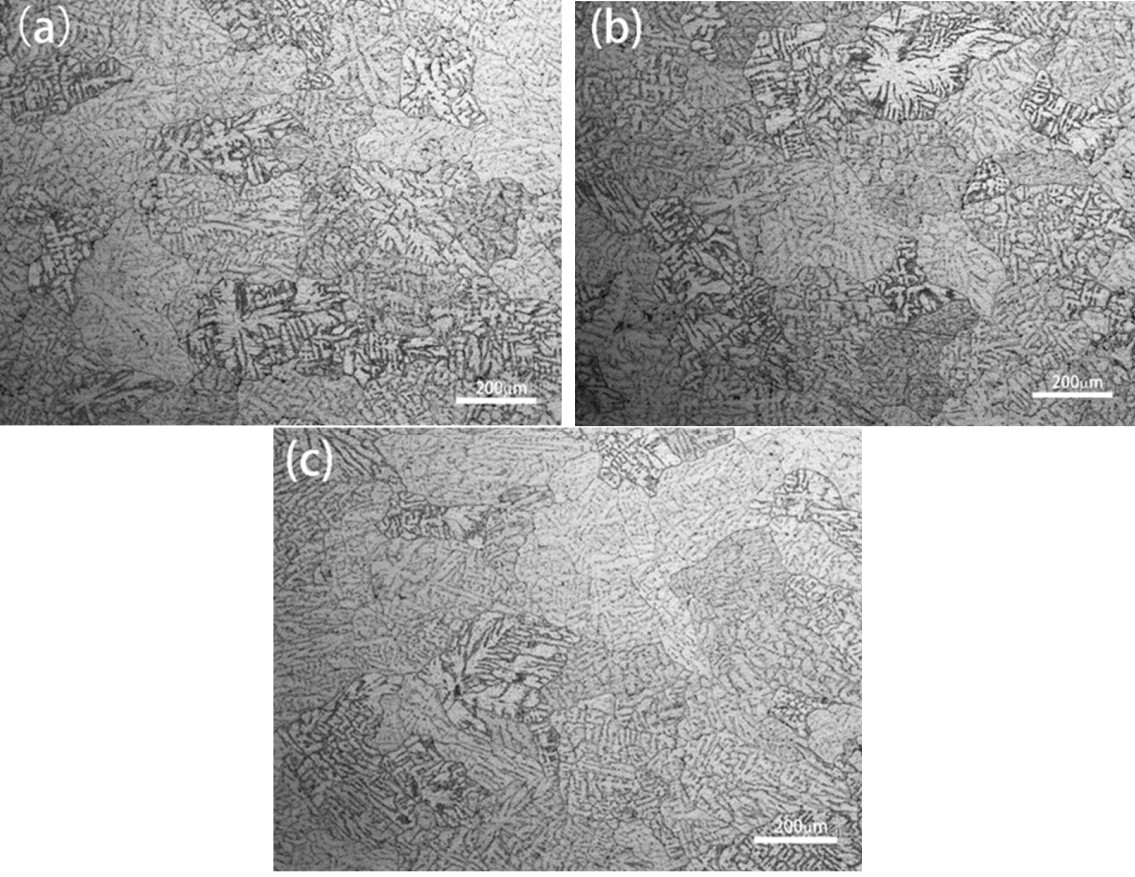

**Figure 1.** The initial microstructure of the studied alloy with (**a**) Zn/Mg ratio of 6.3, (**b**) Zn/Mg ratio of 8.3, and (**c**) Zn/Mg ratio of 10.8.

There are five different temperature conditions, four different strain rates, and three different alloys. In total, 60 curves need to be present. For the readers' convenience, the true stress–strain curves for the alloy with Zn/Mg = 6.3 were chosen to demonstrate the data process routine, while the results of other alloys are calculated using the same methodology.

Figure 2 shows the true stress–strain curves during the hot-compression testing of alloy AA7003 with a Zn/Mg ratio of 6.3 under the strain rate condition of (a) $0.01 \text{ s}^{-1}$, (b) $0.1 \text{ s}^{-1}$, (c) $1 \text{ s}^{-1}$, and (d) $10 \text{ s}^{-1}$ for different temperatures. The true stress–strain curves need to be corrected as the friction could either leading to inhomogeneous deformation or temperature rising. As a result, the deformed samples start to form a barrel shape. The

detailed friction correction method and temperature correction method can be found from Ebrahimi [9] and Z-P Wan et al. [20]. The authors here use the Ebrahimi method to correct the stress–strain curves; the basic equations are:

$$\frac{P_{ave}}{\sigma} = 8b\frac{R}{H}\left\{ \left[\frac{1}{12} + \left(\frac{H}{R}\right)^2\frac{1}{b^2}\right]^{3/2} - \left(\frac{H}{R}\right)^3\frac{1}{b^3} - \frac{m}{24\sqrt{3}}\frac{\exp(-b/2)}{\exp(-b/2)-1} \right\} \quad (1)$$

$$b = \frac{4m/\sqrt{3}}{(R/H) + \left(2m/3\sqrt{3}\right)} \quad (2)$$

$$m = \frac{(R/H)b}{\left(4/\sqrt{3}\right) - \left(2b/3\sqrt{3}\right)} \quad (3)$$

where $m$ is the constant friction factor in the compression test, $\sigma$ is the corrected true stress, Pave is the uncorrected external pressure applied to specimens in compression (the measured stress), $b$ is the barrel parameter, and $R$ and $H$ are the radius and height of samples during compression. Temperature correction was also carried out so the raw data and corrected data are shown in Figure 2. It is shown in Figure 2a that, for a given temperature condition, the true stress values initially increase rapidly with increasing true strain values. It reaches its maximum values after a small number of strain values and remains constant for the rest of the strain values before it descends to tremendous strain values. Comparing stress values at different temperatures demonstrates that the alloy is more resistant to deformation at lower temperatures. This trend is generally observed in other strain rate conditions, as shown in Figure 2b–d. When comparing different strain rate conditions for a given temperature, the true stress increases with increasing strain rate conditions (as shown in the same color in different figures). It is also interesting to find out that the true stress is relatively stable at low-strain-rate conditions (as shown in Figure 2a,b). The strain–stress curves become more fluctuated at relatively high-strain-rate conditions (as shown in Figure 2c,d). Moreover, the stress curves exhibit a significant drop at high-strain-rate conditions. This could be an indication of the instability of the studied alloys.

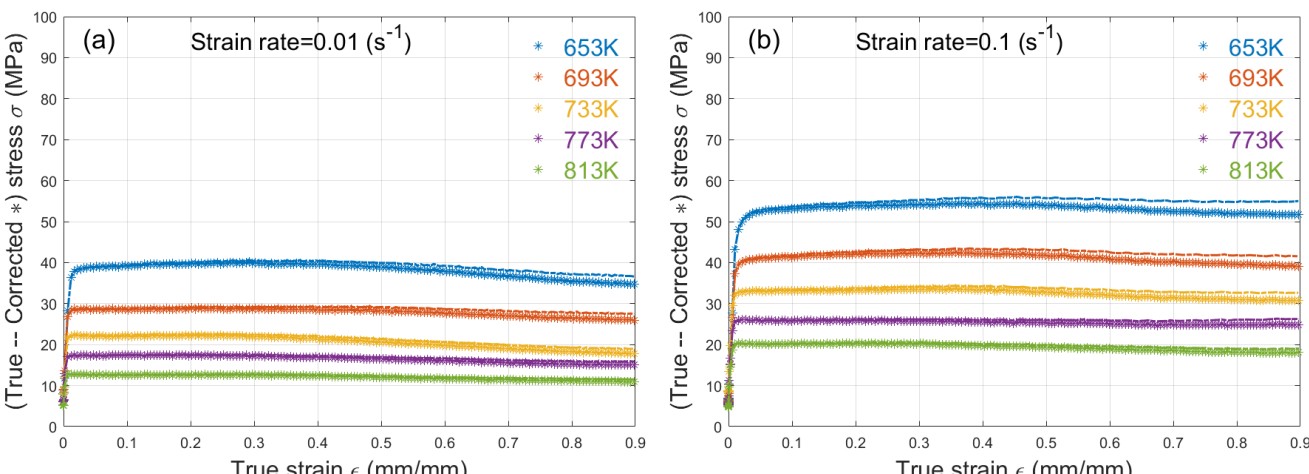

**Figure 2.** *Cont.*

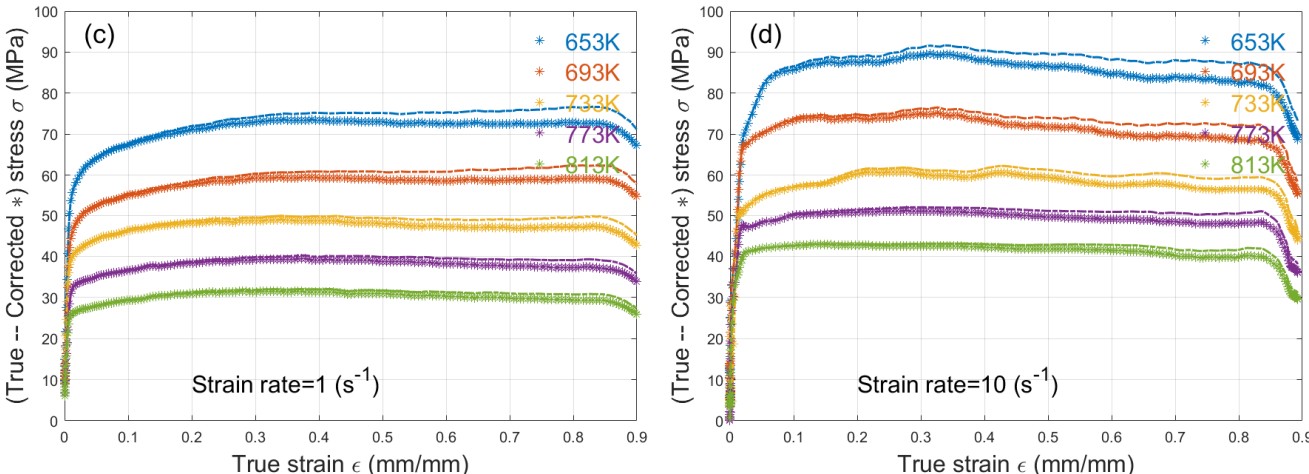

**Figure 2.** Raw and corrected (friction and temperature) stress vs. true strain curves of alloy AA7003 with Zn/Mg ratio of 6.3 during hot-compression testing at different strain rates for different temperatures. (**a**) strain rate of 0.01 s$^{-1}$, (**b**) strain rate of 0.1 s$^{-1}$, (**c**) strain rate of 1 s$^{-1}$ and (**d**) strain rate of 10 s$^{-1}$.

Table 2 summarizes the detailed flow stress values (in MPa) of alloy with a Zn/Mg ratio of 6.3 at different temperatures and strain rates for various strains.

**Table 2.** Flow stress values (in MPa) of alloy with Zn/Mg ratio of 6.3 at different temperatures and strain rates for various strains.

| Strain | Strain Rate, s$^{-1}$ | Temperature (K) | | | | |
|---|---|---|---|---|---|---|
| | | 653 | 693 | 733 | 773 | 813 |
| 0.2 | 0.01 | 39.73 | 28.90 | 22.16 | 17.42 | 12.55 |
| | 0.1 | 53.86 | 42.07 | 33.35 | 25.86 | 20.28 |
| | 1 | 71.13 | 57.64 | 48.13 | 38.34 | 30.98 |
| | 10 | 87.47 | 73.90 | 60.46 | 50.75 | 42.62 |
| 0.4 | 0.01 | 39.54 | 28.46 | 21.26 | 16.93 | 12.31 |
| | 0.1 | 54.23 | 42.22 | 33.43 | 25.62 | 19.63 |
| | 1 | 73.42 | 59.38 | 48.63 | 39.05 | 31.34 |
| | 10 | 87.91 | 73.14 | 60.25 | 50.46 | 42.17 |
| 0.6 | 0.01 | 37.91 | 27.58 | 19.83 | 16.08 | 11.69 |
| | 0.1 | 53.22 | 40.74 | 31.95 | 24.98 | 18.93 |
| | 1 | 72.62 | 58.57 | 47.28 | 38.24 | 30.27 |
| | 10 | 84.68 | 70.09 | 57.51 | 49.02 | 41.49 |
| 0.8 | 0.01 | 35.55 | 26.34 | 18.25 | 15.16 | 11.23 |
| | 0.1 | 51.92 | 39.65 | 30.99 | 24.86 | 18.11 |
| | 1 | 72.76 | 59.11 | 47.05 | 37.28 | 29.36 |
| | 10 | 82.38 | 68.74 | 56.43 | 48.04 | 39.94 |

### 3.2. Constitutive Equations

The constitutive equations describe the relationship between flow stress and hot-working parameters, such as strain, deformed temperatures, and strain rates. Sellars and McTegart developed a hyperbolic sine model in 1960 [24]. This model has been widely used to describe the workability of different alloys. Workability depends on the initiated microstructures, chemical compositions, and processing histories. For example, the annealed materials (O temper) exhibit better workabilities than the deformed materials (H temper). It should also be noted that the friction between the sample's edges and the dies, such as adiabatic heating during the deformation, may significantly influence the true stress–true strain curves. This could be corrected by introducing lubricant during the

deformation process or empirically corrected by estimation [25,26]. Since we used graphite paste on both ends of the samples, the friction effect can be negligible in this case.

In the 1960s, Sellars and McTegart proposed that the isothermal stress–strain relation is based on Arrhenius equations [24]:

$$\dot{\varepsilon} = f(\sigma) \exp\left(-\frac{Q}{RT}\right) \tag{4}$$

where the strain rate unit is $s^{-1}$; $R$ is gas constant, $Q$ is activation energy for hot deformation, unit in kJ/mol; T is isothermal temperature, unit in Kelvin; and $f(\sigma)$ is the strain-related equation, also called Zener–Hollomon parameters in some publications.

It turns out that the alloy may exhibit different deformation mechanisms within other strain regions. Therefore, many publications proposed describing the data using different subfunctions for different intervals:

$$\text{When } \alpha\sigma < 0.8, \ \dot{\varepsilon} = A_1 \sigma^{n_1} \exp\left(-\frac{Q}{RT}\right) \tag{5}$$

$$\text{When } \alpha\sigma > 1.2, \ \dot{\varepsilon} = A_2 \exp(\beta\sigma) \exp\left(-\frac{Q}{RT}\right) \tag{6}$$

$$\text{For all other } \alpha\sigma, \ \dot{\varepsilon} = A[\sin h(\alpha\sigma)]^n \exp\left(-\frac{Q}{RT}\right) \tag{7}$$

where $n_1$, $\beta$, and $\alpha$ are the material's constants, and $\alpha = \beta/n_1$. Taking the natural logarithm on both sides of the equation yields:

$$\ln\dot{\varepsilon} = \ln A_1 + n_1 \ln\sigma - Q/RT \tag{8}$$

$$\ln\dot{\varepsilon} = \ln A_2 + \beta\sigma - Q/RT \tag{9}$$

$$\ln\dot{\varepsilon} = \ln A + n\ln[\sin h(\alpha\sigma)] - Q/RT \tag{10}$$

According to Equation (5), the slope value of the linear relationship between $\ln\dot{\varepsilon}$ and $\ln(\sigma)$ is the $n_1$ value. According to Equation (6) at different temperatures, the slope value of the linear relationship between $\ln\dot{\varepsilon}$ and $\sigma$ is $\beta$. The $\alpha$ can be calculated accordingly. The detailed plots are shown in Figure 3a–c. The activation energy $Q$ value for hot deformation can be extrapolated by linear fitting for $1/T$ and $\ln[\sin h(\alpha\sigma)]$ at a given strain rate condition.

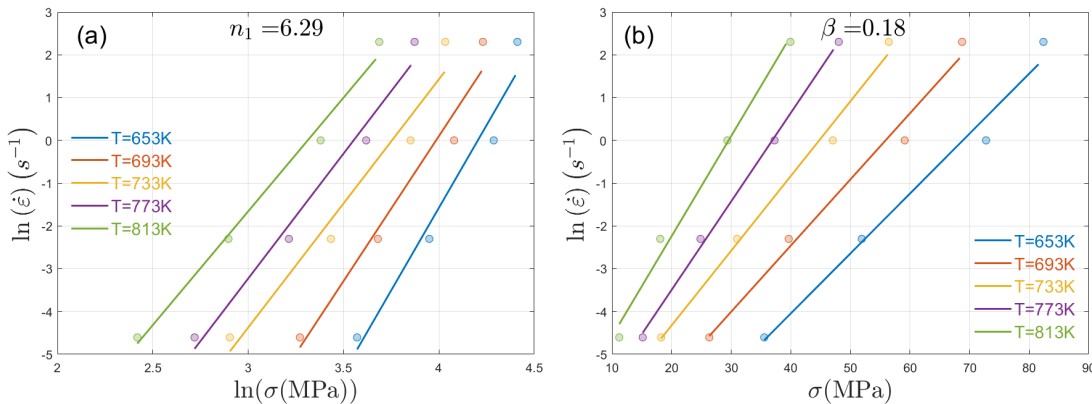

**Figure 3.** *Cont.*

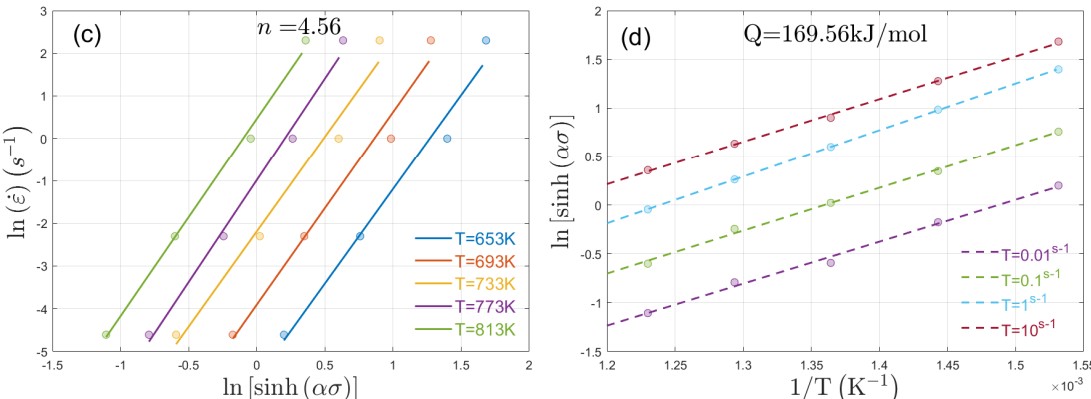

**Figure 3.** (**a**) linear fitting for $\ln(\sigma)$ and $\ln(\dot{\varepsilon})$ at different temperatures; (**b**) linear fitting for $\sigma$ and $\ln(\dot{\varepsilon})$ at different temperatures; (**c**) linear fitting for $\ln[\sin h(\alpha\sigma)]$ and $\ln(\dot{\varepsilon})$; and (**d**) linear fitting for $1/T$ and $\ln[\sin h(\alpha\sigma)]$ at given strain rate conditions.

Applying this methodology to other strain conditions, one can calculate the activation energy Q values for studied alloys (as shown in Figure 4). It was demonstrated that an alloy with a Zn/Mg ratio of 10.8 exhibited deficient activation energy compared to the other two studied alloys. This could be an indication that this alloy is easy to be deformed. The activation energy for the other two alloys with close Zn/Mg ratios (Zn/Mg ratio of 6.3 and Zn/Mg ratio of 8.3) demonstrated very similar values. However, their values show a different trend with increasing strain.

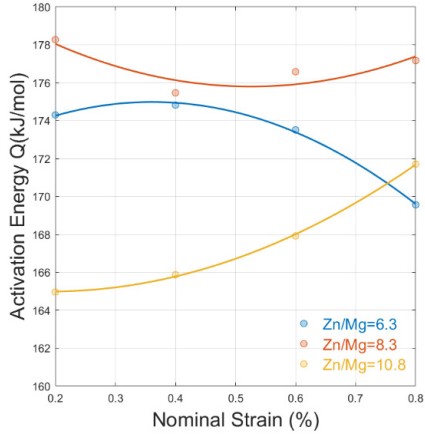

**Figure 4.** Calculated Activation energy value under different nominal strain conditions for three studied alloys with different Zn/Mg ratios.

### 3.3. Processing Maps

The constitutive equation may be helpful in the interpretation of strain–stress curves. However, the processing map could be a straightforward method to describe the workability of the studied alloys. Although it is an explicit representation of the response of studied alloys, it has been widely applied in many process parameter selections.

According to Prasad and Srivatsana [27,28], the input energy causing deformation at a given temperature could be dissipated by heat or the so-called "conduction entropy" and microstructural changes induced by dislocation movement.

$$P = G + J = \int_0^{\dot{\varepsilon}} \sigma d\dot{\varepsilon} + \int_0^{\sigma} \dot{\varepsilon} d\sigma \tag{11}$$

where the first integral (G content) is dissipated energy as temperature arises, while the second integral (J co-content) is energy dissipated due to microstructural changes.

$$m = \frac{\partial J}{\partial G} = \frac{\dot{\varepsilon}\partial\sigma}{\sigma\partial\dot{\varepsilon}} = \frac{\partial \ln\sigma}{\partial \ln\dot{\varepsilon}} \tag{12}$$

The strain rate sensitivity ($m$) is given by Equation (9). This strain rate sensitivity value defines the relationship between $\ln\dot{\varepsilon}$ and $\ln\sigma$. According to Prasad [4,5,8], the $m$ value is generally between 0 and 1 for aluminium alloys. Equation (8) states that:

$$\Delta J/\Delta P = \frac{m}{m+1} \tag{13}$$

The efficiency of power dissipation ($\eta$) is, therefore, defined by:

$$\frac{\Delta J/\Delta P}{(\Delta J/\Delta P)_{linear}} = \frac{2m}{m+1} = \eta \tag{14}$$

The denominator in Equation (11) indicates that the G-content is equal to J co-content in the ideal dissipation system. The efficiency of power dissipation ($\eta$) describes how close the current system is compared to the ideal dissipation system, since the J-content is more related to the microstructural changes. Therefore, the efficiency of power dissipation ($\eta$) essentially describes the microscopic deformation mechanism of the materials within the range of applied temperatures and strains. The efficiency of power dissipation ($\eta$) changes with temperature and strain rate to form a power dissipation map, representing the microstructure change in the studied materials. Since various failures (such as void formation and cracking propagation) or metallurgical changes (such as dynamic recovery, dynamic recrystallization, etc.) in the plastic deformation process dissipate input energy, with the help of microstructural characterization, the power dissipation diagram can be used to analyze different deformation mechanisms under other deformation conditions. It is necessary, first, to determine the processing instability zone of the studied alloys. According to Prasad [5,8,29], the instability criteria are given by:

$$\xi(\dot{\varepsilon}) = \frac{\partial \ln\left(\frac{m}{m+1}\right)}{\partial \ln\dot{\varepsilon}} + m < 0 \tag{15}$$

Figure 5 shows the constructed processing maps for the studied alloy with different Zn/Mg ratios at different strain conditions. The processing map is an instability diagram overlapped with an energy dissipation diagram at the current strain condition. As shown in Figure 5, the yellow shaded region represents the instability regions. When comparing with other publications, we have presented a series of processing maps that show a systematic change with different conditions. In general, the instability regions for studied alloys are within the lower-temperature and higher-strain-rate conditions. This is consistent with the idea that alloy AA7003 is easier to be deformed when compared to other high-strength 7xxx series alloys [5,9,26]. It is interesting that the studied alloys represent different workability with different Zn/Mg ratios, i.e., different rows in Figure 5. The alloys with a lower Zn/Mg ratio exhibit more significant instability regions. Therefore, it is concluded that alloy AA7003 with a higher Zn/Mg ratio could have better formability than lower Zn/Mg ratios. When comparing different strain conditions, i.e., different columns, it is clear that the higher-strain condition exhibits more significant instability regions. This also agrees with our shared knowledge that higher-strain conditions could lead to void formation and cracking propagation. The result also agrees with Figure 4 that the calculated activation energy value for the alloy with a Zn/Mg ratio of 10.8 is significantly lower than the other two alloys.

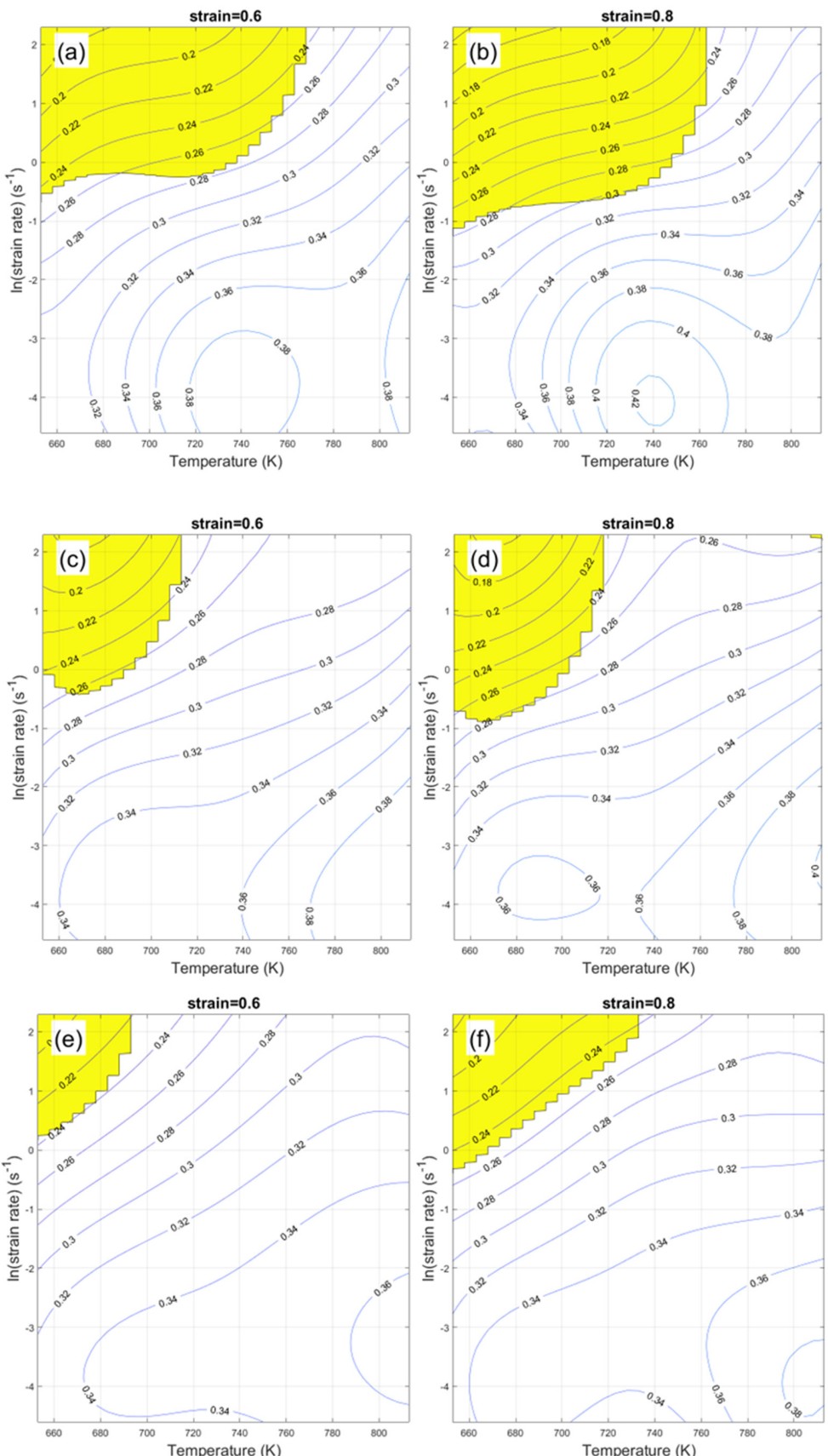

**Figure 5.** The processing maps for alloys with (**a**,**b**) Zn/Mg = 6.3, (**c**,**d**) Zn/Mg = 8.3, (**e**,**f**) Zn/Mg = 10.8, at strain of 0.6 (left column) and 0.8 (right column), respectively, while the shaded region represents the instability regions.

The contour lines in Figure 5 represent the efficiency of power dissipation ($\eta$) at the current strain condition. As discussed earlier, the maximum efficiency is 100%, while G-content is equal to J co-content in the ideal dissipation system. In the current dissipation system, the maximum efficiency ($\eta$) is about 50%, and the high efficiency of power dissipation ($\eta$) is always shown in the lower-right corner of each diagram. This indicates that the input energy is more likely to be dissipated at higher temperatures and low-strain-rate conditions without cracking or instability. Therefore, it is concluded that optimized deformation parameters for ternary alloy AA7003 are within a temperature range of 653–813 K and with strain rates lower than $0.3 \, \text{S}^{-1}$.

## 4. Discussions

As discussed earlier, the J co-content is energy dissipated due to microstructural changes. The alloy with a Zn/Mg ratio of 8.3 was chosen for typical microstructure characterization. The electron backscatter diffraction (EBSD) technique was selected to analyze deformed microstructures, recrystallized microstructures, and substructures quantitively. The specific microstructural characterization is shown in Figure 6. When the grain's misorientation angle ($\theta c$) exceeds 15°, it is classified as a deformed microstructure. Grains consisting of subgrains whose internal misorientation is below 15°, but whose misorientation from subgrain to subgrain is above 2°, are classified as substructures. All the remaining grains are classified as recrystallized. In Figure 6, the colored maps present different microstructures, i.e., blue stands for recrystallized structures, yellow stands for substructures, and red stands for deformed structures. It is shown in Figure 6a that the primary remaining microstructures are deformed structures that coexist with a small number of substructures and few recrystallized structures. When deformed at a higher temperature, i.e., 733 K in Figure 6b, it is shown that more substructures and more recrystallized structures were found. It is shown in Figure 5b that the primary microstructures are substructures. Additionally, the recrystallized structures are significantly increased. Figure 6c shows that when deformed at 813 K, the recrystallized structures become the dominated microstructures.

The statistical analysis of EBSD mappings at a different temperature and at a strain rate of $0.1 \, \text{s}^{-1}$ is shown in Figure 7. It is demonstrated in Figure 7a that the area fraction of recrystallized grains significantly increased with deformation temperatures, while the frequency of deformed microstructures declined dramatically. This indicates that recrystallization occurs rapidly for the alloy with Zn/Mg = 6.3. The frequency of recrystallized structures showed a moderate decrease for the alloy with Zn/Mg = 8.3. However, it is shown in Figure 7b that the substructures arose significantly at different temperature conditions. For the alloy that contained the highest Zn/Mg ratio (as shown in Figure 7c), the substructures exhibited a remarkable frequency at a temperature of 733 K and 813 K. It is also shown that the amount of recrystallized structures increased slightly with temperature. It also should be noticed that the deformed microstructures decreased with rising temperatures within all three alloys. In general, the EBSD results show that the alloy with Zn/Mg = 10.8 exhibited a significantly small amount of recrystallization, while the other two alloys exhibited a moderate amount of recrystallization. This could be an indication that the dynamic recrystallization is the main factor to dissipate the deformation energy. It is also shown that the alloy with Zn/Mg = 10.8 retained a large number of substructures. This could also be an indication that this alloy can be further deformed without cracking.

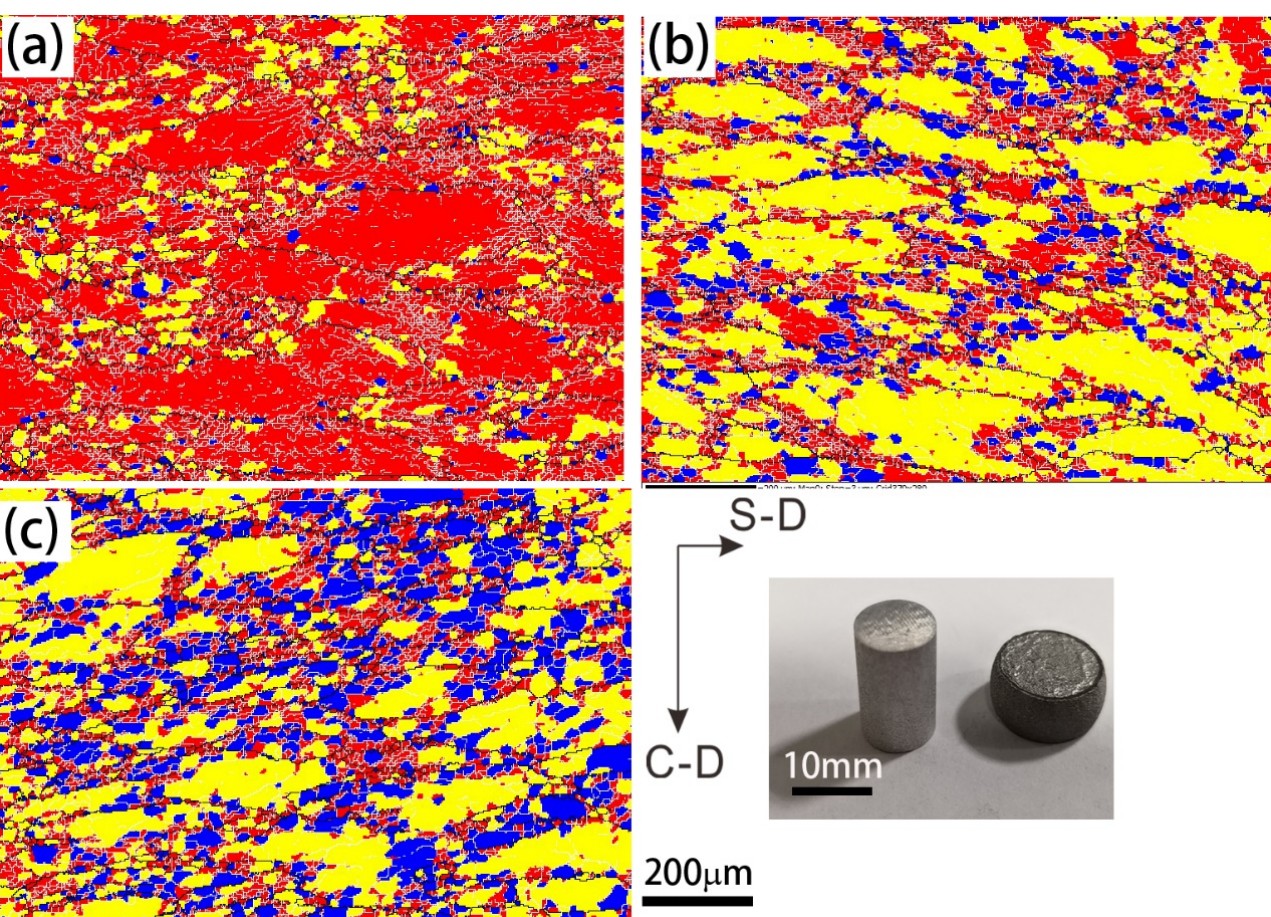

**Figure 6.** The typical EBSD mappings for the alloy with Zn/Mg = 6.3 deformed at (**a**) 653 K, (**b**) 733 K and (**c**) 813 K at a strain rate of $0.1\,\mathrm{S^{-1}}$, with the different colors representing different microstructures, i.e., yellow stand for substructures, blue stand for recrystallized structures and red stand for deformed structures; the bottom right image shows the compressed samples before and after deformation.

Many reports focus on the effects of dispersoid particles [30,31]. It is proven that these particles effectively inhibit the dislocation movement and grain boundary movements [32–35]. In the current research work, ~0.2% Zr was added to the studied alloys. The typical particles are known to be $Al_3Zr$ dispersoids. It is shown in Figure 8a that there are a significant number of spherical $Al_3Zr$ particles; they have an $L1_2$ crystalline structure (as shown by the selected area diffraction pattern (SADP)) and are confirmed to be coherent with the Al matrix. The SADP also indicates that these particles exhibit a simple cubic/cubic orientation relationship with the Al matrix, where the diffraction pattern from $Al_3Zr$ particles is located at {100} planes.

It is shown in Figure 8a that the size of $Al_3Zr$ dispersoids is within the range of 20–50 nm. In fact, due to the same homogenization treatment and Zr addition, all three studied alloys have a similar size distribution of $Al_3Zr$ dispersoids. However, it should be noted that these dispersoids are heterogeneously distributed within grains. Some regions (as shown in Figure 8b) have very-low-number density while others have relatively high-number density (as shown in Figure 8a). Interestingly, these particles constantly interact with dislocations or grain boundaries. It is shown in Figure 8b that the dislocations bypass a single spherical $Al_3Zr$ dispersoid by bowing around. According to classical deformation theory, an Orowan loop is left afterward. The classical deformation theory considers the shear stress or line tension caused by dispersoids themselves. However, it should be noted that the studied alloys exhibit different thermal–mechanical behaviors, given they contain a similar amount of Zr addition. Therefore, it is concluded that the main alloying content,

such as Zn/Mg ratios, could also affect the dynamic recrystallization process. In the current study, about the reaction with recrystallizations, three scenarios can be discerned: the low-Zn/Mg-ratio alloy being deformed with difficulty can cause recrystallization quickly, the medium-Zn/Mg-ratio alloy has a moderate trend of recrystallization, and the high-Zn/Mg-ratio alloy being deformed easily can lead to very low recrystallization. The underlying mechanism is not yet fully understood. More detailed work will be conducted in the future.

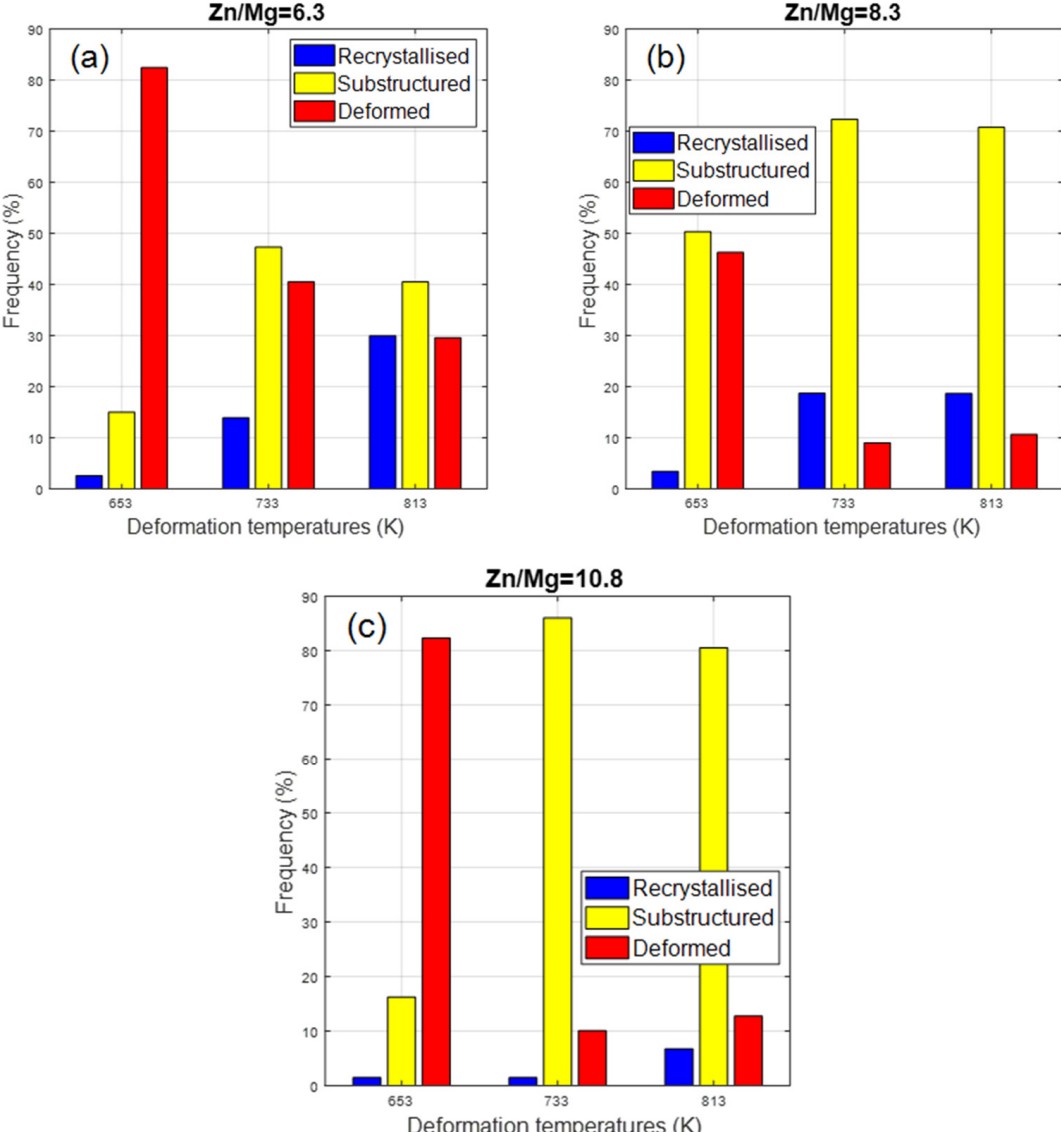

**Figure 7.** Statistical analysis of different area fractions of different microstructures at various deformation temperatures for (**a**) alloy with Zn/Mg = 6.3, (**b**) alloy with Zn/Mg = 8.3, and (**c**) alloy with Zn/Mg = 10.8.

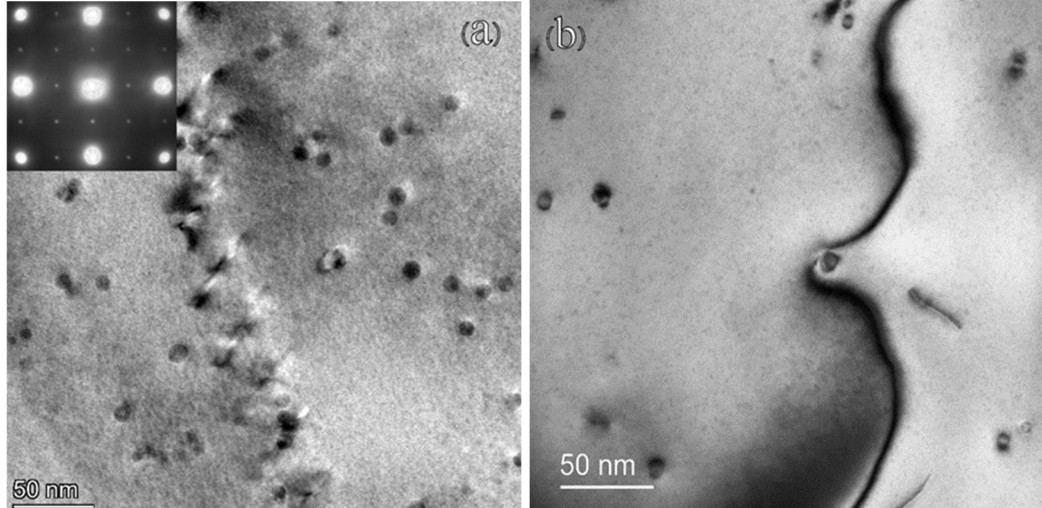

**Figure 8.** TEM observation of the alloy with Zn/Mg = 6.3 deformed at 733 K at a strain rate of 0.1 S$^{-1}$, (**a**) Distribution of Al$_3$Zr dispersoids within grains (an alloy with Zn/Mg = 6.3), viewing from {100}$_{Al}$, (**b**) an Al$_3$Zr dispersoid tangled with dislocation movement during deformation.

## 5. Conclusions

The hot-deformation behavior of three Al-Zn-Mg alloys with different Zn/Mg ratios was studied. The main conclusions are:

(1)  When comparing the processing maps for AA7003 alloys with different Zn/Mg ratios, alloys with a low Zn/Mg ratio of 6.3 led to a problematic hot-deformation capability. In contrast, alloys with a higher Zn/Mg ratio of 10.8 exhibited better workability than lower Zn/Mg ratios.

(2)  The optimized deformation parameters for ternary alloy AA7003 were within a temperature range of 653–813 K and at strain rates lower than 0.3 S$^{-1}$.

(3)  When comparing the microstructures after hot deformation, alloy AA7003 with a lower Zn/Mg ratio of 6.3 had a more negligible fraction of substructures but higher frequency of recrystallized structures. In comparison, the alloy with a higher Zn/Mg ratio of 10.8 had a high fraction of substructures and low frequency of recrystallization.

(4)  The Al$_3$Zr dispersoids were effective in inhibiting the recrystallization for alloy AA7003; three scenarios can be discerned when considering the interaction between dispersoids and recrystallization: the low-Zn/Mg-ratio alloy being deformed with difficulty can cause recrystallization easily, the medium-Zn/Mg-ratio alloy has a moderate trend of recrystallization, and the high-Zn/Mg-ratio alloy contains a minor fraction of recrystallization and, therefore, leads to easy deformability.

**Author Contributions:** Writing—original draft preparation, X.Z.; Writing—review and editing, J.T.; Materials supplier, L.W.; Data collection, Y.Z. (Yan Zhao) and C.J.; Project administration, Y.Z. (Yong Zhang). All authors have read and agreed to the published version of the manuscript.

**Funding:** This research was funded by the National Key Research and Development Program of China (No. 2016YFB0300901) and Guangxi Science & Technology Program (Guike AA22068075).

**Data Availability Statement:** Data is contained within the article.

**Acknowledgments:** The authors would like to thank Foshan Sanshui Fenglu Aluminium Co., Ltd. China, for providing the materials.

**Conflicts of Interest:** The authors declare no conflict of interest.

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
