# Peer review of "Hot Deformation Behavior of Alloy AA7003 with Different Zn/Mg Ratios"

_metals, doi:10.3390/met12091452_

Round 1

Reviewer 1 Report (Previous Reviewer 1)

The authors of the paper "Hot deformation behavior of alloy AA7003 with different Zn/Mg ratios" have answered the part of previous comments. However, some points of the manuscript are questionable. Some parts of the manuscript are needed to be modified accordingly following comments:

1.                 Despite using the graphite foil between dies and samples’ edges the friction effect is significant (please see the barrel shape of the deformed sample in figure 6 due to the friction). At the same time, the authors did not consider the effect of adiabatic heating during the deformation. Stress-strain curves should be corrected for these effects (please, see methods of the correction in 10.1016/j.jallcom.2018.08.010, 10.1134/S0031918X14080031).

2.                 The authors have provided the initial microstructure of the investigated alloys before deformation. It is recommended to give the value of the average grain size for each alloy.

3.                 It is absolutely unclear, how did the authors obtain fracture surfaces during compression. Accordingly stress-strain curves and the appearance of the sample in Figure 6, the fracturing does not proceed during the tests. Are the authors sure that the provided fracture surface images were obtained for investigated samples?

Author Response

The authors of the paper "Hot deformation behavior of alloy AA7003 with different Zn/Mg ratios" have answered the part of previous comments. However, some points of the manuscript are questionable. Some parts of the manuscript are needed to be modified accordingly following comments:

1. Despite using the graphite foil between dies and samples’ edges the friction effect is significant (please see the barrel shape of the deformed sample in figure 6 due to the friction). At the same time, the authors did not consider the effect of adiabatic heating during the deformation. Stress-strain curves should be corrected for these effects (please, see methods of the correction in 10.1016/j.jallcom.2018.08.010, 10.1134/S0031918X14080031).

We appreciate the reviewer’s suggestion, and we have corrected the stress-strain curves based on Ebrahimi’s method. So that all the figures related to the hot-deformation maps have been recalculated. It seems that the changes are limited, but the results are more convincing.

2. The authors have provided the initial microstructure of the investigated alloys before deformation. It is recommended to give the value of the average grain size for each alloy.

We have measured the grain size of the initial microstructure, and updated the values.

3. It is absolutely unclear, how did the authors obtain fracture surfaces during compression. Accordingly stress-strain curves and the appearance of the sample in Figure 6, the fracturing does not proceed during the tests. Are the authors sure that the provided fracture surface images were obtained for investigated samples?

We choose samples that have small cracks, and teared apart, so that the fracture surface can be observed. We also clarified this method within the manuscript.

Reviewer 2 Report (New Reviewer)

The article concerns medium-strength Al-Zn-Mg alloys, which are precipitation hardening alloys. These types of alloys are a very popular material, especially in automotive and construction applications. The authors presented the results of deformation tests at different strain rates and various temperature. On the basis of the obtained research results, they developed maps of treatment conditions for the AA7003 alloy with different Zn / Mg 61 ratios. During the research, the authors changed the Zn / Mg ratios and assessed the natural aging effect and deformation capacity of these alloys. They presented many different research methods, incl. isothermal compression stress tests performed on a Gleeble thermo-simulation machine and used SEM and TEM techniques to evaluate microstructures.

 I have a few comments:

1. Fig. 1 - no information about the magnification - please complete

2. Fig. 9. - no information about the magnification - please complete

3. The authors presented studies in which ~ 0.2% Zr was added to the tested alloys. What was the purpose of adding Zr to alloys? Why was Zr chosen? Why such an amount of Zr?

4. Do the adopted strain rates correlate with the extrusion conditions of these alloys? What is the actual rate of deformation when extruding such alloys (industrial conditions)?

Author Response

The article concerns medium-strength Al-Zn-Mg alloys, which are precipitation hardening alloys. These types of alloys are a very popular material, especially in automotive and construction applications. The authors presented the results of deformation tests at different strain rates and various temperature. On the basis of the obtained research results, they developed maps of treatment conditions for the AA7003 alloy with different Zn / Mg 61 ratios. During the research, the authors changed the Zn / Mg ratios and assessed the natural aging effect and deformation capacity of these alloys. They presented many different research methods, incl. isothermal compression stress tests performed on a Gleeble thermo-simulation machine and used SEM and TEM techniques to evaluate microstructures.

 I have a few comments:

1. 1 - no information about the magnification - please complete

The figures embed with scale-bar, so there is no need for magnifications.

2. 9. - no information about the magnification - please complete

The figures embed with scale-bar, so there is no need for magnifications.

3. The authors presented studies in which ~ 0.2% Zr was added to the tested alloys. What was the purpose of adding Zr to alloys? Why was Zr chosen? Why such an amount of Zr?

This is a very good question, first, the Zr was chosen because it has a significant effect on inhibit recrystallisation and instability during deformation. Second, alloy AA7003 is a commercially registered alloy, its Zr content is within the range of 0.05-0.25 (wt%). We chose ~ 0.2% Zr so that the alloy can be deformed without too much recrystallisation and instability during deformation process.

4. Do the adopted strain rates correlate with the extrusion conditions of these alloys? What is the actual rate of deformation when extruding such alloys (industrial conditions)?

The purpose of this article is to show a systematically change of processing maps for alloy AA7003 with different Zn/Mg ratios. The results can be indicative for industrial applications, for example, according to our research, the alloy with high Zn/Mg ratio can be deformed easily or on faster extrusion conditions than the other two alloys. The industrial applications are quite complex, but we would recommend temperatures and strain rates combination conditions on an average basis.

Reviewer 3 Report (New Reviewer)

In the article Hot deformation behavior of alloy AA7003 with different Zn/Mg ratios, the effect of variation of the Zn/Mg components on the deformation processes of alloys is considered. In general, this area of ​​research is very important not only from a scientific point of view, but also from a practical one, since the data obtained make it possible to contribute to the development of methods for controlling deformation processes in materials. The article corresponds to the subject of the declared journal and can be accepted for publication after the authors answer a number of questions that the reviewer has during a detailed analysis of the data presented.

1. In the introduction, the authors should give more information about the object of study, its prospects and the possibilities of applying the results obtained in practice.

2. The authors should explain what exactly causes the variation of the Zn/Mg components in the alloy, as well as the effect of this variation on the phase composition of the alloy.

3. In the experimental part, the authors should give more information about the testing conditions and the selected ranges of external influences and annealing. It should be explained what exactly determines the choice of temperatures and what conditions of technological processes they correspond to.

4. The results of the EBSD mapping data require additional explanations, as well as a description of how exactly the contents of various fractions in the composition of the alloys were determined.

5. The data of mechanical tests require additions in the field of explanation of the observed effects depending on the exposure temperature. What exactly causes the increase in stability with increasing temperature, and whether these effects are associated with partial relaxation of defects.

Author Response

In the article Hot deformation behavior of alloy AA7003 with different Zn/Mg ratios, the effect of variation of the Zn/Mg components on the deformation processes of alloys is considered. In general, this area of ​​research is very important not only from a scientific point of view, but also from a practical one, since the data obtained make it possible to contribute to the development of methods for controlling deformation processes in materials. The article corresponds to the subject of the declared journal and can be accepted for publication after the authors answer a number of questions that the reviewer has during a detailed analysis of the data presented.

1. In the introduction, the authors should give more information about the object of study, its prospects and the possibilities of applying the results obtained in practice.

Response: We add more background information on why we are conducting this research. The industrial always wants to extrude the materials as fast as possible. But they are facing cracking or unstable for a given composition. It is therefore interesting to explore the effect of different Zn/Mg ratios on the workability of variable alloy compositions.

2. The authors should explain what exactly causes the variation of the Zn/Mg components in the alloy, as well as the effect of this variation on the phase composition of the alloy.

Response: This is quite a big question. In fact, the whole article explores the different Zn/Mg ratios on the workability of studied alloys. The results show that alloys with a higher Zn/Mg ratio of 10.8 exhibit better workability than lower Zn/Mg ratios. For one thing, alloys with a lower Zn/Mg ratio exhibit more significant instability regions. For another, the area fraction of recrystallized grains is significantly increased with deformation temperatures within alloy with a lower Zn/Mg ratio. The underline mechanism of how exactly the variation of the Zn/Mg components causes the differences apparently need more analysis on both deformation behaviors and microstructures characterization. But that would be too much for the current article.

3. In the experimental part, the authors should give more information about the testing conditions and the selected ranges of external influences and annealing. It should be explained what exactly determines the choice of temperatures and what conditions of technological processes they correspond to.

Response: We have detailed explained the experimental conditions. The homogenisation was carried out at 733K for 48 hours. The isothermal deformation compression tests were carried out on a computer servo-controlled Gleeble-1500 thermo-simulation machine. All samples were lubricant with graphite paste at both ends to reduce friction and increase thermal conductivity. The samples were heated to target temperatures at a constant heating rate of 1 K/s and held at setting temperatures for 5 min before compression. Five different temperatures (653, 693,733,773, and 813K) were chosen to cover the real industrial manufacturing situations.

4. The results of the EBSD mapping data require additional explanations, as well as a description of how exactly the contents of various fractions in the composition of the alloys were determined.

Response: More explanation has been added in EBSD part.

5. The data of mechanical tests require additions in the field of explanation of the observed effects depending on the exposure temperature. What exactly causes the increase in stability with increasing temperature, and whether these effects are associated with partial relaxation of defects.

Response: We choose samples that have small cracks, and teared apart, so that the fracture surface can be observed. We also clarified this method within the manuscript.

Round 2

Reviewer 1 Report (Previous Reviewer 1)

The authors have answered previous comments. Minor corrections are required:

1. References numbers in line 128 are incorrect.

2. "Friction" should be instead of friction.

3. It is not possible to obtain images of the fracture surfaces for the presented deformation modes. The fracturing of aluminum alloy hard to proceed during the deformation at 0.1 1/s. The part about the fracturing should be removed from the manuscript.

4. Title of the part 4.1 is incorrect. 

Author Response

The authors have answered previous comments. Minor corrections are required:

  1. References numbers in line 128 are incorrect.

The reference number has been updated.

  1. "Friction" should be instead of friction.

The word “fiction” in Figure 2 caption has been changed to “friction”.

  1. It is not possible to obtain images of the fracture surfaces for the presented deformation modes. The fracturing of aluminum alloy hard to proceed during the deformation at 0.1 1/s. The part about the fracturing should be removed from the manuscript.

The fracture surfaces of figure 8 and related content have been removed.

  1. Title of the part 4.1 is incorrect. 

Title 4.1 have been changed to “discussion”.

Reviewer 3 Report (New Reviewer)

The authors answered all the questions, the article can be accepted for publication.

Author Response

The authors answered all the questions, the article can be accepted for publication.

We appreciate reviewer’s suggestions.

This manuscript is a resubmission of an earlier submission. The following is a list of the peer review reports and author responses from that submission.

Round 1

Reviewer 1 Report

In the paper "Hot deformation behavior of alloy AA7003 with different Zn/Mg ratios", the authors have investigated the influence of the Zn and Mg content on the effective activation energy, flow instability, and dissipation energy during hot deformation of the Al-Zn-Mg alloy. The authors have shown that the alloy with a higher Zn/Mg ratio has better deformability. The presented results seem to be interesting. However, some parts of the manuscript are needed to be modified accordingly following comments:

1.                 The part of the cited references is too old. It is recommended to consider the ore last references devoted to the hot deformation and processing mapping of the Al-Zn-Mg alloys (e.g., 10.1016/j.jallcom.2022.163690, 10.3390/app11104587, etc).

2.                 The friction between the sample’s edges and the dies such as adiabatic heating during the deformation may significantly influence the true stress – true strain curves. Did the authors consider this fact?

3.                 The initial microstructure of the investigated alloys before deformation should be added to the manuscript. Was the grain size of the alloys the same for all materials? The difference in grain size may significantly influence the hot deformation behaviour.

4.                 Obtained values of the dissipation energy percentage are too high. Usual values for aluminium alloys are not higher than 0.5. For the Al–6.2Zn–0.70Mg–0.3Mn–0.17Zr alloy (10.1016/j.jallcom.2015.01.228) the dissipation energy percentage is in the range of 10 – 35 %. It seems that the authors did not correctly calculate the value of Æž. It should be twice lower. The authors should provide a detailed method of the processing maps construction (approximations, interpolation functions, etc.). It is recommended to recalculate processing maps and compare the results with similar processing maps obtained by other scholars.

5.                 The authors did not provide any microstructural evidence of the flow instability at low temperatures and high strain rates (cracks, pores, etc.) It is recommended to provide additional microstructural investigations of the samples deformed in the unstable region.

6.                 It is unclear, what state of the alloy was chosen for the investigation by TEM. Additional information should be added to the Figure 7 caption.

7.                 Minor corrections:

-                     The temperature units should be unified through the manuscript.

-                     The error is in eq. (1.11).

Reviewer 2 Report

Congrat with a fine paper - only thing is a final proof reading to ensure the use of right english word - it is a little confusing with the use of Kelvin versus Celsius - i could suggest to use Celsius, however, still stating the use of absolute temperatur in equations

Reviewer 3 Report

The manuscript is focused on the study of workability at high temperature of Al-Zn-Mg alloys with three different Zn/Mg ratio, through the mechanical compressive analysis, from which is calculated the activation energy, and the morphological analysis.

The novelty of this article is limited as well as the overall design of the research, that is restricted to mechanical compressive tests at high temperature of the alloys. The scientific contribute is thus few relevant and for these reasons this manuscript is not suitable for the publication on this journal.